# Single-Sided Portable NMR Investigation to Assess and Monitor Cleaning Action of PVA-Borax Hydrogel in Travertine and Lecce Stone

**DOI:** 10.3390/molecules26123697

**Published:** 2021-06-17

**Authors:** Valeria Stagno, Chiara Genova, Nicole Zoratto, Gabriele Favero, Silvia Capuani

**Affiliations:** 1Earth Sciences Department, Sapienza University of Rome, Piazzale Aldo Moro 5, 00185 Rome, Italy; valeria.stagno@uniroma1.it; 2National Research Council Institute for Complex Systems (CNR-ISC) c/o Physics Department, Sapienza University of Rome, Piazzale Aldo Moro 5, 00185 Rome, Italy; silvia.capuani@isc.cnr.it; 3Department of Chemistry and Drug Technologies, Sapienza University of Rome, Piazzale Aldo Moro 5, 00185 Rome, Italy; nicole.zoratto@uniroma1.it

**Keywords:** NMR, relaxation time *T*_2_, single-sided NMR, hydrogel, cleaning, Paraloid B72

## Abstract

In this work, we investigated the potential of PVA-borax hydrogel for cleaning limestones and the dependence of the cleaning on the porosity of the rock and on the action time of the hydrogel treatment. Towards this goal, we used a nuclear magnetic resonance (NMR) spectrometer, developed for non-invasive and non-destructive applications on cultural heritage. *T*_2_-NMR parameters were quantified on different samples of Lecce stone and Travertine cut perpendicular (Pe) and parallel (Pa) to the bedding planes under different experimental conditions: untreated samples, treated with Paraloid B72 and cleaned with PVA-PEO-borax hydrogel applied for 4 min and 2 h. The *T*_2_ results suggest that the effectiveness of the cleaning strongly depended on the porosity of the stones. In Lecce stone, the hydrogel seemed to eliminate both the paramagnetic impurities (in equal measure with 4 min and 2 h treatment) and Paraloid B72. In Travertine Pe, characterized by a smaller pore size compared to Lecce stone, no significant effects were found regarding both the cleaning and the treatment with Paraloid B72. In Travertine Pa, characterized by a larger pore size than the other two samples, the hydrogel seemed to clean the paramagnetic agents (it worked better if applied for a longer time) but it did not appear to have any effect on Paraloid B72 removal.

## 1. Introduction

Stone represents one of the most widely employed materials for the realization of cultural heritage artefacts, including statues, fountains, and buildings.

Among them, Travertine and Lecce stone are calcareous lithotypes mainly composed of calcium carbonate (CaCO_3_) and are commonly employed all over the Mediterranean area, particularly in the edification of Italian historical buildings. These lithotypes are characterized by a similar chemical composition but different macro and micro-structures and physical properties [1].

Stones are porous media, characterized by a high surface-to-volume ratio [2]. Gas and liquid can diffuse from the environment into these materials, producing degradation phenomena that may compromise, not only the natural physiological aging, but also the interactions occurring between the stone matrix and external factors that lead to irreversible chemical, physical, and aesthetical alterations to the materials. Concerning outdoor exposed stone artefacts, it is very common to observe deterioration phenomena caused by the deposition of various substances on the exposed surfaces, such as dust, pollutants, metal ions, airborne particles, or the microorganisms variously dispersed in the atmosphere. In most cases, the deterioration mechanisms are activated by water. For example, water allows the solubilization of the atmospheric sulfur dioxide (SO_2_) present in polluted environments, which reacts with calcium carbonate (CaCO_3_) to form gypsum (CaSO_4_·2(H_2_O)) [3,4], a less noble material, and one that is much more friable and absorbent than the original carbonate surface [5]. These newly formed gypsum incrustations retain various aero-dispersed particles of pollutants and organic materials, giving rise to the so-called black crusts: dark areas (due to soot particles) covered by an irregular, dendrite-like hard crust [6], which also represents one of the most studied and relevant problems characteristic of outdoor preserved carbonate artistic materials. The formation of gypsum on the stone surface is a rapid process and can also be accelerated by the deposition of particulate matter rich in metals and metal oxides, which can act as a catalyst in the sulphation reaction [7]. Moreover, the black crust shows some differences in microstructure and porosity compared to the substrate, leading to the detachment of the black crust itself and the gradual weakening of the stone of the monument surface [8,9]. Water, when retained inside the porous matrix, also allows the proliferation of photosynthetic microorganisms (algae, cyanobacteria, and lichens), pioneers in the biological succession, followed by chemotropic microorganisms (fungi and bacteria), which, together, are responsible for the biodeterioration that causes aesthetic, chemical, and physical damage to the biocolonized stones (i.e., discolorations, biopitting, surface roughness, fractures, and detachments) [10,11].

Other significant modifications to artistic surfaces can be induced by old restoration and coating interventions, which involved the application of various synthetic and natural polymeric materials [12]. They have been used in the past as coatings, protectives, consolidants, adhesives, etc., with the idea of enhancing the brightness and the visual properties of the surfaces and preserving the structure from natural decay. However, it has been widely demonstrated that most of these substances lead to an overall impairment of artwork conservation status, causing significant alterations to the appearance (mainly a yellowing caused by the photooxidation and the natural aging of these protectives) and to the physicochemical properties of the original artefact (i.e., porosity, capillarity, vapor permeability, and surface wettability) [12]. Moreover, the presence of organic materials employed as coatings (for example natural and synthetic waxes, acrylic and siloxane resins, perfluoro-polyethers, fluorinated polyolefin, and fluoro-elastomers [13]) inevitably promotes the proliferation of microflora, contributing to the previously cited biodeterioration processes [14].

Among the acrylic polymers, Paraloid B72, an ethyl methacrylate/methyl acrylate (EM/MA) copolymer employed as a coating, consolidant, and adhesive, is a widely used commercial product, especially for stone materials [15]. It is also used as a water-repellent on outdoor exposed surfaces to limit the physicochemical and biological problems related to the presence of water inside the porous matrix and on surfaces [13]. Its frequent employment depends on its transparency, reversibility, absence of color, stability, resistance to deterioration, solubility in acetone and ethanol, and ease of application [16]. However, some drawbacks related to Paraloid B72 have been noted. The most common problems can be ascribed to the reduction in the permeability of the treated surfaces, but more generally, to the physicochemical variations of the stone matrix, as well as the chemical alterations of the polymer due to cross-linking reactions and chain scissions, causing a loss of solubility and the consequent irreversibility of the application [17,18]. Moreover, studies based on nuclear magnetic resonance (NMR) relaxometry and imaging investigations showed how Paraloid B72 is inappropriate for travertine since, by decreasing the average diameter of the pores, it facilitates the entry of water by capillary rise, which does not get released, as the resin prevents evaporation [19,20]. 

In this scenario, the need is evident to have products for cleaning stone surfaces that can eliminate potential degradation products and preserve cultural heritage for future generations. The cleaning procedure is one of the most delicate and potentially harmful operations for the product. In order to improve this procedure and its safety, high-performance cleaning systems are mandatory [21,22,23].

In this context, soft matters, including micelle, microemulsions, gels, and gel-like systems are receiving great attention for the cleaning of many different cultural heritage materials [21,22,23,24,25,26]. These new generation substances have shown, as their main advantages, good cleaning properties, high selectivity, low toxicity, and low environmental impact [26,27]. Among these products, highly promising are those based on Poly (vinyl alcohol) (PVA), a versatile water-soluble polymer characterized by biocompatible, biodegradable, and bioinert properties [28], and which is able to react with borax (sodium tetraborate salt) through a di-diol condensation to form completely transparent highly viscous aqueous fluids, also called highly viscous polymeric dispersions (HVPDs), and characterized by a 3D network [29]. On the other hand, to choose the best product to clean specific artistic stonework, it is essential to use a non-invasive and non-destructive investigation technique that allows studying the effectiveness of the cleaning products while monitoring of the artwork conservation status.

In this work, a non-destructive portable NMR instrument was used to test the potential cleaning action of an HVPD called PVA-borax hydrogel (or PVA-based hydrogel), in Travertine and Lecce stone. In the last two decades, the application of ^1^H-NMR [30] to cultural heritage porous stones has increased thanks to the development of protocols based on longitudinal (*T*_1_) and transversal (*T*_2_), relaxation time measurement [31,32,33]. Capitani et al. 2012 [34] suggested that a breakthrough has certainly been the development of portable single-sided (or unilateral) NMR sensors [35,36,37], through which liquid in porous materials of any size can be monitored in a non-destructive and non-invasive modality to obtain structural information.

The hypotheses this work tested were: (1) NMR relaxation time measurements obtained from the surface to a 2-mm depth of stone samples can provide information about PVA-borax hydrogel’s potential for removing Paraloid B72; (2) the cleaning action of the PVA-borax hydrogel depends on the porous feature of stones, as well as on the application time needed to get good cleaning results. 

## 2. Results

In Figure 1, the *T*_2_ relaxation time distribution for the PVA based hydrogel (Figure 1a) and for the Paraloid/acetone solution (Figure 1b) are displayed. Since the main constituent of the hydrogel is water, its NMR signal was strong, showing two predominant *T*_2_ components around 561 ms and 166 ms. While a continuous distribution between these two major peaks is observable, minor peaks can be found at 45, 15, and 1.6 ms. The *T*_2_ distribution obtained for the Paraloid/acetone solution is characterized by a longer *T*_2_ = 14 ms and two very short *T*_2_ of 0.73 ms and 0.27 ms.

Figure 2 shows the *T*_2_ distributions of the three untreated samples: Lecce stone, Travertine cut perpendicular to the bedding planes (Travertine Pe), and parallel to the bedding planes (Travertine Pa). Water present in the pores of all the samples showed three *T*_2_ components. Among them, Travertine Pa (red curve) had the shortest *T*_2_ values, equal to 5, 0.59, and 0.18 ms. Lecce stone (light blue curve) shows three *T*_2_ peak-areas around 10, 0.94, and 0.27 ms, whereas Travertine Pe (green curve) is characterized by *T*_2_ = 65, 0.89, and 0.21 ms. 

In Figure 3, the *T*_2_ distribution after PVA-based gel application for 4 min and 2 h on each untreated sample is displayed. Note that after the PVA-gel application, the relaxation times appear modified for each sample compared to those measured on untreated surface samples. In particular, Lecce stone (Figure 3a) and Travertine Pa (Figure 3c) show a greater *T*_2_ change after both application times (for 4 min and 2 h) of the PVA-based gel. Overall, a general shift towards higher *T*_2_ values after the gel treatment can be observed in all the samples.

In Figure 4, the *T*_2_ changes in the three samples after the Paraloid B72 treatment can be observed. For the two Travertine stones, PB72 provided a small increment of the *T*_2_, whereas for Lecce stone the *T*_2_-related peaks shifted towards shorter *T*_2_ values. 

Figure 5 displays the result of the *T*_2_ NMR measurement, monitoring the gel cleaning procedure in order to remove the PB72 layer. For Lecce stone (Figure 5a) the gel application produced an increase of the *T*_2_ values compared to those measured with the PB72 layer on the sample surface. *T*_2_ = 0.22 ms components were measured after both the application times of the PVA-based gel. When the PVA gel was left for 4 min, other *T*_2_ peaks were detected, whereas when it was left for 2 h only another *T*_2_ peak was measured. In this latter case, the *T*_2_ peak coincided with the intermediate *T*_2_ peak detected for the untreated Lecce stone. Travertine Pe showed changes of *T*_2_ during the treatment. For this sample, when the gel application time was 4 min a general increment of the *T*_2_ values compared to those measured in both the untreated sample and the PB72 treated sample was found. The only exception was the highest peak, which seemed to decrease after the gel application. Conversely, when the gel was applied for 2 h the two shorter *T*_2_ components coincided with those measured in the untreated sample. Travertine Pa showed a similar behavior, with only two *T*_2_ peaks detected after 4 min of gel application and a general shift of the *T*_2_ towards higher values after the gel applications.

## 3. Discussion

In recent years, the employment of PVA based hydrogels crosslinked with borax has been spreading in the field of cleaning of cultural heritage materials [38]. The success of these products is attributable to some of their intrinsic characteristics, in particular, (i) the atoxicity in their chemical composition and their being harmless towards the environment and human beings; (ii) their ability to incorporate organic solvents; and (iii) the ease of their application and removal procedures [29,38,39]. However, in the literature, there are only few papers investigating the effectiveness of PVA-hydrogels for cleaning rocks [5,40,41,42,43]. Recently, Riedo et al. [44] tested the effectiveness of PVA-borax hydrogel in generic limestone samples treated with Paraloid B72 and cleaned using a 4-min gel application time. Inspired by the Riedo et al. [44] paper, we investigated samples of specific limestones, Travertine Pe, Travertine Pa, and Lecce stone, characterized by different porosities, and we studied the effectiveness of PVA-borax hydrogel, selecting two different gel application times: 4 min and 2 h. Towards this goal, we quantified the NMR proton *T*_2_ relaxation times of water (and/or other molecules) by wetting the pores of the stone samples using a portable NMR spectrometer, specifically developed for applications in the cultural heritage field. The spin dephasing quantified by *T*_2_ is a measure of the mobility of water molecules, which in turn is indicative of the pore size in a porous system. When no, or very poor, paramagnetic substances are present and the pores are completely filled with water, there is a direct correlation between the *T*_2_ values extracted from the decay of the NMR signal and the pore size. The smaller the pore, the shorter the relaxation time, because of the high probability of interaction with the surface [31]. If the pores are wet and not soaked in water, the contribution of the surface dephasing of the proton spins becomes more important, which in turn is closely linked to the characteristics of the pore surface (more or less wettable) [45,46,47]. The surface *T*_2_ is inversely related to the surface-to-volume-ratio (S/V) of pores [48,49].

Considering this general behavior of the *T*_2_ parameter, the results of non-destructive NMR monitoring of treated and non-treated limestones samples are discussed in the following sections.

### 3.1. NMR Characterization of the PVA-Based Hydrogel and Paraloid/Acetone Solution

In a homogeneous sample, the *T*_2_ depends on the mobility of the protons. In free and pure water at room temperature, it measures about 3 s. As the molecular weight increases, and therefore the molecular dynamics decrease, the *T*_2_ value decreases. In this context, the two *T*_2_ components detected for the PVA-based hydrogel (Figure 1a), characterized by *T*_2_ = 561 ms and *T*_2_ = 166 ms, can be associated with water that is free or slightly interacting with other large polymers (i.e., PVA and PEO), with also a possible exchange between the two water populations. The other three *T*_2_ components are associated with three different proton populations of polymers. The two lower *T*_2_ components of the Paraloid/acetone solution (Figure 1b) can be attributed to protons of acetone solvent interacting with Paraloid, which is a large molecule with PM = 100,000, while the *T*_2_ = 14 ms component can be related to the dissolved Paraloid B72 or less interactive protons in acetone. 

### 3.2. NMR Characterization of Travertines and Lecce Stone

In Figure 2, the different behavior for the transversal relaxation time among the Lecce, Travertino Pa, and Travertino Pe can be attributed to their porous structure differences. Due to the inverse relationship existing between *T*_2_ and the surface-to-volume ratio (S/V) of the pores in a porous system [50], our results can provide information about the different porosities of our three samples. The three main *T*_2_ components detected in each limestone suggest three possible ranges of pores: small (short *T*_2_), medium (intermediate *T*_2_), and large (long *T*_2_).

Overall, Travertine Pa showed a shorter *T*_2_ than Travertine Pe, suggesting a greater S/V of its pores. This result is in agreement with the literature [50], where Travertine Pa (i.e., cut parallel to the bedding plane) was described as having larger surface pores, characterized by a wider surface area, than those of Travertine Pe (i.e., cut perpendicular to the bedding plane). The longest *T*_2_ of the Lecce stone, which can be associated with larger pores, is characterized by an intermediate behavior between the two kinds of Travertine. This suggests that the largest pores of Lecce stone are smaller than those of Travertine Pe but larger than those of Travertine Pa. Moreover, Lecce stone has the largest pores for the range of the small pores. These results are in general agreement with the literature [31,50,51,52,53,54,55], showing the potential of portable single sided NMR for probing the pore features of different limestones.

### 3.3. NMR Assessment of the PVA-Gel Application on an Untreated (no PB72) Stones Surface

On the basis of the discussions in Section 3.2, the hydrogel–stone matrix interaction can be evaluated. Since the hydrogel is mainly composed of water, the predominant interaction between the gel system and the stone porous matrix is the water adsorption, but also the gel penetration inside the open porosity of the stone surface due to the pressure exerted during the modeling of the gel. Both the phenomena depend on the characteristic of stone porosity. The plots in Figure 3 suggest that the gel application mainly affected the *T*_2_ of Lecce stone and Travertine Pa, and in particular the *T*_2_ component associated with larger pores. In general, the average value of all *T*_2_ components increases after gel application and removal. Furthermore, in Travertine Pa T_2_ increased significantly when the treatment time increased. This behavior suggests that PVA-borax hydrogel was able to remove dust or dirt made up of paramagnetic ions and molecules from the sensitive volume of the samples investigated with an NMR profiler. In fact, *T*_2_ decreased in value in parallel with the paramagnetic substance concentration. In this test, Travertine Pa benefited from a longer gel treatment application time, while Travertine Pe, which is characterized by smaller pore size and porosity compared to the other limestones, seemed to be unaffected by the potential cleaning action of the gel. 

### 3.4. NMR Monitoring of the PVA-Gel Cleaning of PB72 from Stones Surface

The effect of the Paraloid B72 coating on the *T*_2_ (Figure 4) suggests that the three studied samples had different behaviors towards this substance. Again, Travertine Pe seemed unaffected by the Paraloid B72 coating. Conversely, the Lecce stone after the PB72 coating exhibited one main *T*_2_ component at a shorter *T*_2_ compared to the *T*_2_ values of the untreated sample. This result suggests that the PB72 was well absorbed by the surface of the Lecce stone, going deep into the stone itself. This led to a decrease in the size of the largest pores (the average *T*_2_ decreases) and therefore a lower dispersion in the size of the pores (only one main *T*_2_ component). PB72 affects the longer *T*_2_ component of Travertine Pa, increasing its mean value compared to that of the untreated sample. This behavior is explained by the fact that the large pores are filled with PB72 (also seen visually), thus exhibiting approximately the same previously measured longer *T*_2_ value of the PB72 (Figure 1, the *T*_2_ peak component around 10 ms). 

PVA-gel cleaning of PB72 from stones should result in the *T*_2_ returning to the initial values obtained from the untreated surfaces. Figure 5 shows that this happened only for Lecce stone. After the PVA-based hydrogel treatment, Lecce stone again showed all three *T*_2_ components, and quite close to the *T*_2_ components of the untreated sample. This means that the hydrogel had sufficiently eliminated the PB72 absorbed by the Lecce stone, returning a multiplicity of *T*_2_ components that reflected an increased dispersion of the pore size. No conclusions for the Travertine Pe can be reached, as the differences between *T*_2_ components in the cases treated with PB72, untreated, and treated with gel were not significant. An exception was the longest *T*_2_, which was decreased because of the presence of gel residues inside the large pores, as shown in Figure 10a. In Travertine Pa treated with gel compared to the untreated one, both times of gel application (4 min and 2 h) produced an increase in the *T*_2_ values of the two shorter components (i.e., associated with the smaller pores) and, conversely, a reduction in the longer *T*_2_ associated with larger pores. Again, this result may be due to solid gel residues within the large pores, as shown in Figure 10b.

The interaction between polymers, hydrogels, and the porous matrix is very complex because it depends on multiple chemical–physical and topological parameters. However, this was not the aim of this work. In this paper we investigated, with a non-destructive NMR instrument dedicated to cultural heritage applications, the cleaning of PVA-borax hydrogel against PB72 applied to the surface of Lecce and Travertine samples. The used protocol, although based solely on the *T*_2_ relaxation time measurement, suggests that the effectiveness of the cleaning strongly depends on the stone porosity. The Lecce stone was characterized by a distribution of pore size from about 10 to 35 microns [54,55]. In the Lecce stone the PVA-borax hydrogel seemed to eliminate both paramagnetic impurities (in equal measure with 4-min and 2-h treatments) and the PB72. In the Travertine Pe, characterized by a smaller pores size but with a wider and heterogeneous pore distribution compared to the Lecce stone [50,51], the proposed method did not show significant effects due either to the cleaning or the treatment with PB72. Finally, in Travertine Pa, characterized by a larger pore size than the other two samples [50,51], the hydrogel seemed to clean the paramagnetic agents (it worked better if applied for a longer time) but did not seem to have an effect on the PB72 removal.

## 4. Materials and Methods

### 4.1. Materials 

Several stone samples (i.e., limestones) belonging to two different lithotypes, Travertine and Lecce stone, were studied. Concerning the Travertine stones, all the samples were cut from the same Travertine slab. For this reason, they present the same chemical composition, despite differences in the physical properties. One group of Travertine samples was cut perpendicular to the bedding planes (Travertine Pe), while another group was cut parallel to the bedding planes (Travertine Pa). The cut involved a difference in the porosity and capillarity of the Travertine, where the highest pore dimensions, capillarity absorption, and anisotropy was reached parallel to the structure [50,51], while for the perpendicular cut to the bedding planes the pores were smaller and discontinuous [50]. All the samples displayed in Figure 6 were 5 × 5 × 2 cm^3^ in size. 

In general, Travertine is a calcareous stone with 97–99% [52] CaCO_3_ and characterized by a porosity around 10% [51]. It was widely used as a building material during the Roman period, so much so that it is considered “The stone of Rome” [53], but its employment as a building material is still common [51]. On the other hand, Lecce stone is a biocalcarenite [31,54,55] that takes its name from the city of Lecce in the south of Italy. It was especially employed during the Baroque period for buildings and sculptures. Lecce stone shows a calcitic cement in which microfossils with a size of 100–200 µm [54,55] can be found. Moreover, calcium carbonate (CaCO_3_) is its main constituent, around 90–97% [31,54,55]. Lecce stone is characterized by a total porosity around 47% [31,54,55], of which 36% is in the range of mesopores. Nevertheless, only 39% of its porosity is accessible to water [54,55]. During all NMR measurements, the stone samples resided in a climate chamber with a constant temperature of T = 21 ± 1 °C and relative humidity (RH) of 94 ± 3%, reached by using a saturated saline solution of potassium sulphate (K_2_SO_4_). Samples were left to equilibrate with the surrounding environment. 

The HVPD employed in this research was conceived and described in a previous study by Riedo et al. [44]. Its components are poly(vinylalcohol) (87–89% hydrolyzed, Mw 85,000–12,400, Sigma-Aldrich, Milano, Italy), poly(ethyleneoxide) (Mw. 37000–4400, Sigma-Aldrich, Milano, Italy), and sodium tetraborate decahydrate (Sigma-Aldrich, Milano, Italy), while Paraloid B72 in pellets is a commercial product available in many hardware stores and was purchased from Colori e Vernici F.lli Pernesi (Rome, Italy).

### 4.2. Methods

#### 4.2.1. Hydrogel and Paraloid B72 Preparation

A solution containing Paraloid B72 was prepared by dissolving Paraloid B72 pellets in acetone at 2% (*w*/*w*). The pellets were allowed to react with the acetone and were kept in a borosilicate glass bottle for two days. To facilitate the complete dissolution of the pellets, the solution was mixed with a magnetic stirrer without heating. The poly(vinylalcohol) (PVA) based hydrogel, crosslinked with borax and enriched with poly(ethyleneoxide) (PEO), was also prepared following the procedure described in the paper of Riedo et al. [44], obtaining a final product whose composition was 3% PVA, 2% PEO, and 0.6% borax. PEO was introduced into the formulations to improve the mechanical properties of the gels and their compatibility with organic solvents. The above concentrations allowed obtaining a stable product characterized by optical transparency and suitable mechanical properties for the application on different surfaces, as well as the possibility of its removal by a simple peeling. These latter aspects represent the benefits of gels and gel-like systems thanks to their high intrinsic elasticity, which allows their versatile adaptation to the sample shape, which maximizes the contact with the artistic surfaces [39] and permits an easy removal that does not leave any residuals [29,38].

#### 4.2.2. Preliminary Cleaning Tests

Some preliminary tests to determine the best application time of the PVA-based hydrogel for Paraloid B72 removal were carried out. These preliminary tests were performed on a glass Petri dish, a non-porous and non-absorbent surface definitively different to the stone surfaces investigated in this work. The solution containing Paraloid B72 (2% *w*/*w*) dissolved in acetone was applied, with the help of a paint brush, on four different zones of the glass Petri dish. 

Once dried, the four Paraloid stains became opaque (Figure 7a): each stain was treated with the PVA-PEO borax based HVPD (Figure 7b), which was left for 4 min (stain 1), 30 min (stain 2), 2 h (stain 3), and 12 h (stain 4), respectively, after its application. The upper part of the Petri dish was employed to cover the hydrogel, to avoid the complete evaporation of its water, as suggested in the paper of Riedo et al. [44]. As shown in Figure 7c, the exposure times of 4 and 30 min proved to be insufficient for a complete removal of Paraloid from the glass surface. Conversely, better results were obtained in the case of 2 h, where only few traces of Paraloid B72 could be observed. The in visu observations showed that best results occurred in the case of a time exposure corresponding to 12 h, where no traces of Paraloid B72 were observed, although the PVA-PEO borax hydrogel system completely dried, because of the evaporation of the water, which is a not recommended for allowing a complete and quick detachment of the treatments from the substrata.

In virtue of these preliminary observations, all the following applications of the PVA-PEO borax hydrogel on stones were carried out for 4 min and 2 h. These times were chosen because 4 min was the best application time proposed by Riedo et al. [44], whereas, the application time of 2 h seemed a good compromise to increase the action time of the PVA-based gel for PB72 removal and to limit the water evaporation of the gel system, preserving its mechanical properties. 

#### 4.2.3. NMR *T*_2_ Measurements 

NMR relaxometry measurements [30,56,57] were performed by using a BRUKER minispec mq-ProFiler with a single-sided magnet that generates a static magnetic field of 0.4 T. In Figure 8, a schematic representation of how the single-sided magnet functions is displayed. The single-sided NMR instrument [58] with a permanent magnet size of 50 × 50 mm^2^ has a sensitive area containing an RF coil that can be put in contact with the sample surface. This RF coil generates a second magnetic field that produces the hydrogen nuclei excitation in a volume of the sample called the sensitive volume. The sensitive volume is the region of the sample from which the NMR signal comes. As depicted in Figure 8, the intensity of the NMR signal decreases with the depth of the sample surface or with the distance from the RF coil [36,59]. By varying the resonance frequency, thanks to the use of different RF-probes, the penetration depth inside the sample can be changed [59]. In this work, the single-sided spectrometer was equipped with a probe for performing experiments with a 2-mm depth from the sample surface, characterized by a ^1^H-resonance frequency of 17 MHz and dead time of 2 µs.

To monitor each step of the cleaning tests, the transversal relaxation time [30] (*T*_2_) was acquired using a Carr–Purcell–Meiboom–Gill (CPMG) sequence. The optimized acquisition of the used parameters is reported in Table 1, where TR is the repetition time, TE is the echo time, and NS is the number of scans. Each measurement was repeated three times and data were exported and analyzed.

Data were processed in MATLAB (MATLAB R2021a) by using inverse Laplace transform (ILT), which is the most widely used method for *T*_2_ relaxometry data analysis [60], to extract the *T*_2_ distribution and the associated spin-population probability. The mean value and standard deviation (SD) of each *T*_2_ component was evaluated.

Statistical evaluation was made by analysis of variance (ANOVA) to test for differences among groups (*T*_2_ values among three different untreated, treated with PB72, and treated with PVA-based gel limestones). Then, a post hoc analysis, using the Student–Newman–Keuls method, was applied for between group comparisons at a significant level of *p* < 0.05, when a significant difference was detected by ANOVA.

#### 4.2.4. Cleaning Tests on Stones 

In a first step, the *T*_2_ components of the PVA based hydrogel, of the Paraloid/acetone solution, and of the three untreated samples (i.e., Lecce stone, Travertine Pe, and Travertine Pa) were measured. Then, the *T*_2_ distribution was obtained during the cleaning tests on stones. The cleaning tests included both the application of the PVA-PEO borax hydrogel on the untreated surfaces of the stones and on the surfaces of the stones treated with Paraloid B72.

The direct application of the hydrogel on the surface of the stones was done in order to assess the interactions between the stone matrix and the HVPD after 4 min and 2 h of application. To this end, three samples (Lecce stone, Travertine Pe, and Travertine Pa) were cleaned with the PVA-borax hydrogel. In a first phase, the gel was left for 4 min, then it was removed, and the sample was left to dry. After that, the *T*_2_ of the stone was measured. In a second phase the same three samples were covered with the hydrogel that was left for 2 h, again after gel removal and sample drying, NMR measurements were performed.

The second cleaning test involved the use of the Paraloid B72 solution, which was employed to coat one side (surface 5 × 5 cm^2^) of other three stone samples (Lecce stone, Travertine Pe, and Travertine Pa). Three layers of Paraloid were applied with a paint brush until a uniform film on the surface was formed. This was followed by the cleaning procedure that involved the application of the poly(vinylalcohol) (PVA) based hydrogel crosslinked with borax and enriched with poly(ethyleneoxide) (PEO) on the stone samples. The product was applied on the surfaces with a spatula and left for 4 min or 2 h. At the beginning of the application, the gel presented a very viscous appearance, and particular care was required for a uniform application. However, once applied on the surfaces, the PVA-PEO borax hydrogel adapted to the shape of the samples and a homogeneous transparent layer was obtained (Figure 9).

After the application, the treated stones were covered, in order to avoid the complete evaporation of the water composing the PVA-PEO borax hydrogel. The removal of the gel was accomplished by the help of a spatula, which allowed the detachment of the most adherent parts. After a certain drying time, the *T*_2_ of the samples was measured. The three stones surfaces after the gel treatment of 2 h can be observed in Figure 10, which shows the presence of minimal solid gel residuals. 

## Figures and Tables

**Figure 1 molecules-26-03697-f001:**
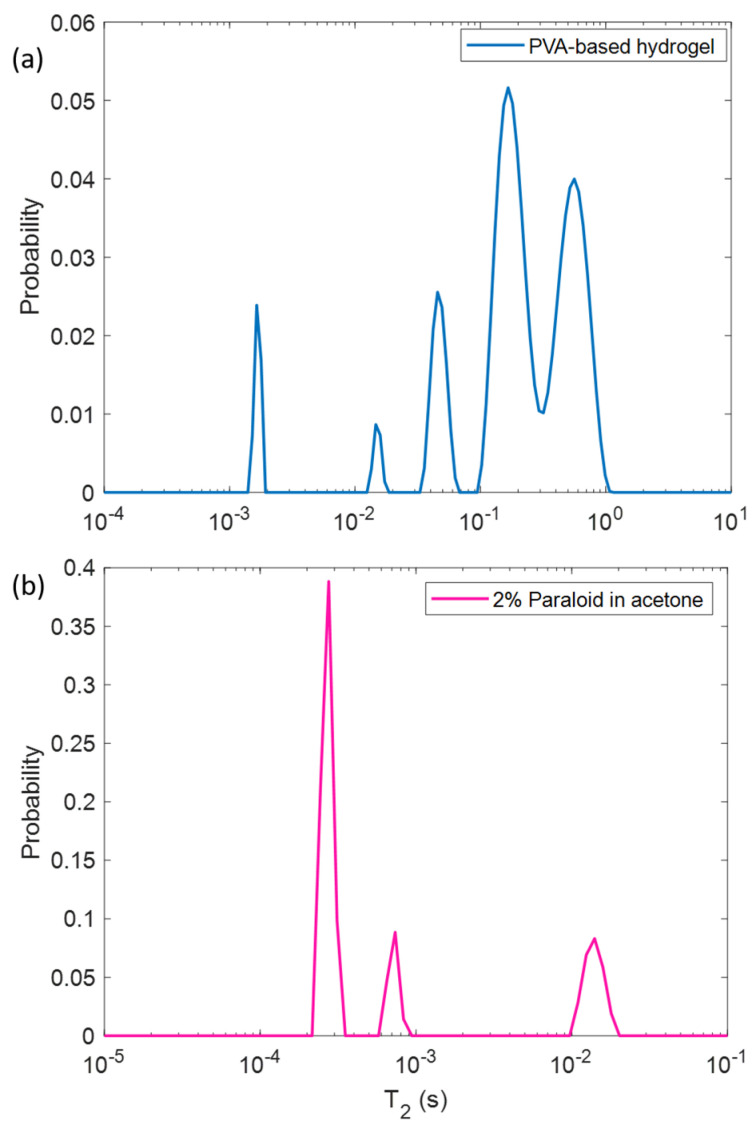
*T*_2_ distribution for (**a**) the PVA-based gel system and (**b**) the Paraloid/acetone solution (2% *w*/*w*).

**Figure 2 molecules-26-03697-f002:**
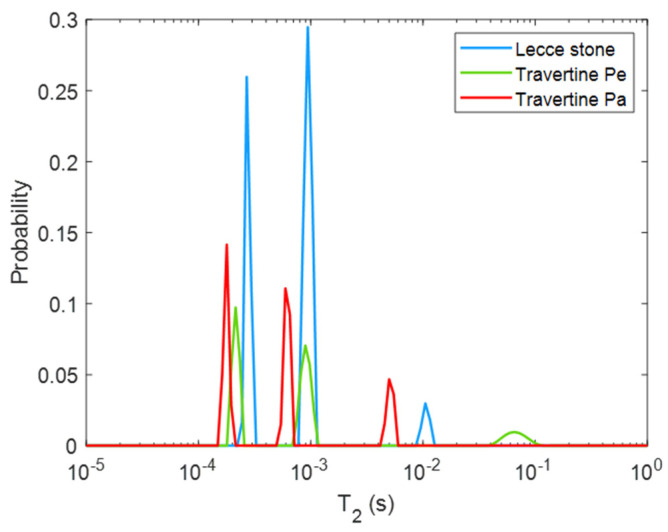
*T*_2_ distribution of water in the three untreated samples.

**Figure 3 molecules-26-03697-f003:**
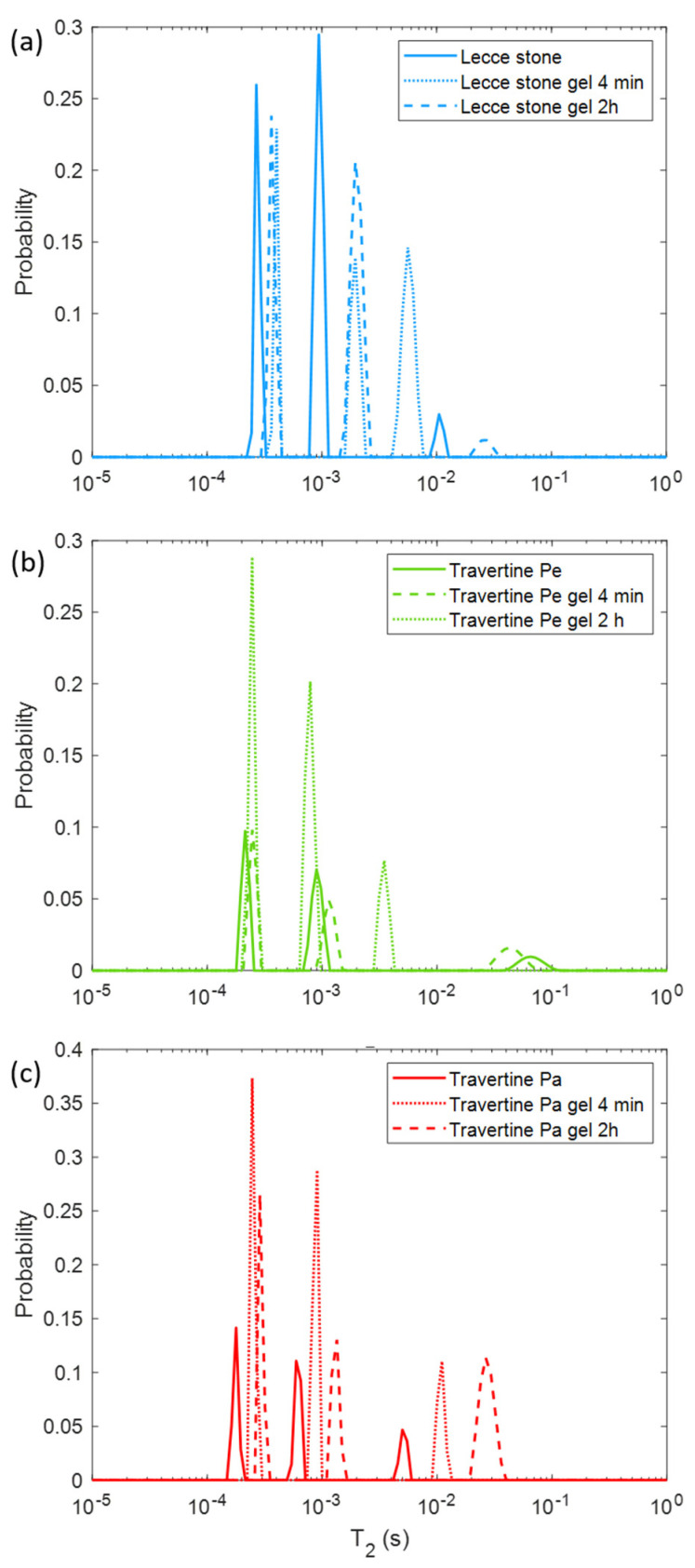
Transversal relaxation time (*T*_2_) distribution for (**a**) Lecce stone, (**b**) Travertine Pe, and (**c**) Travertine Pa before the treatment (solid lines) and after the PVA-borax gel treatment with application times of 4 min (dotted lines) and 2 h (dashed lines).

**Figure 4 molecules-26-03697-f004:**
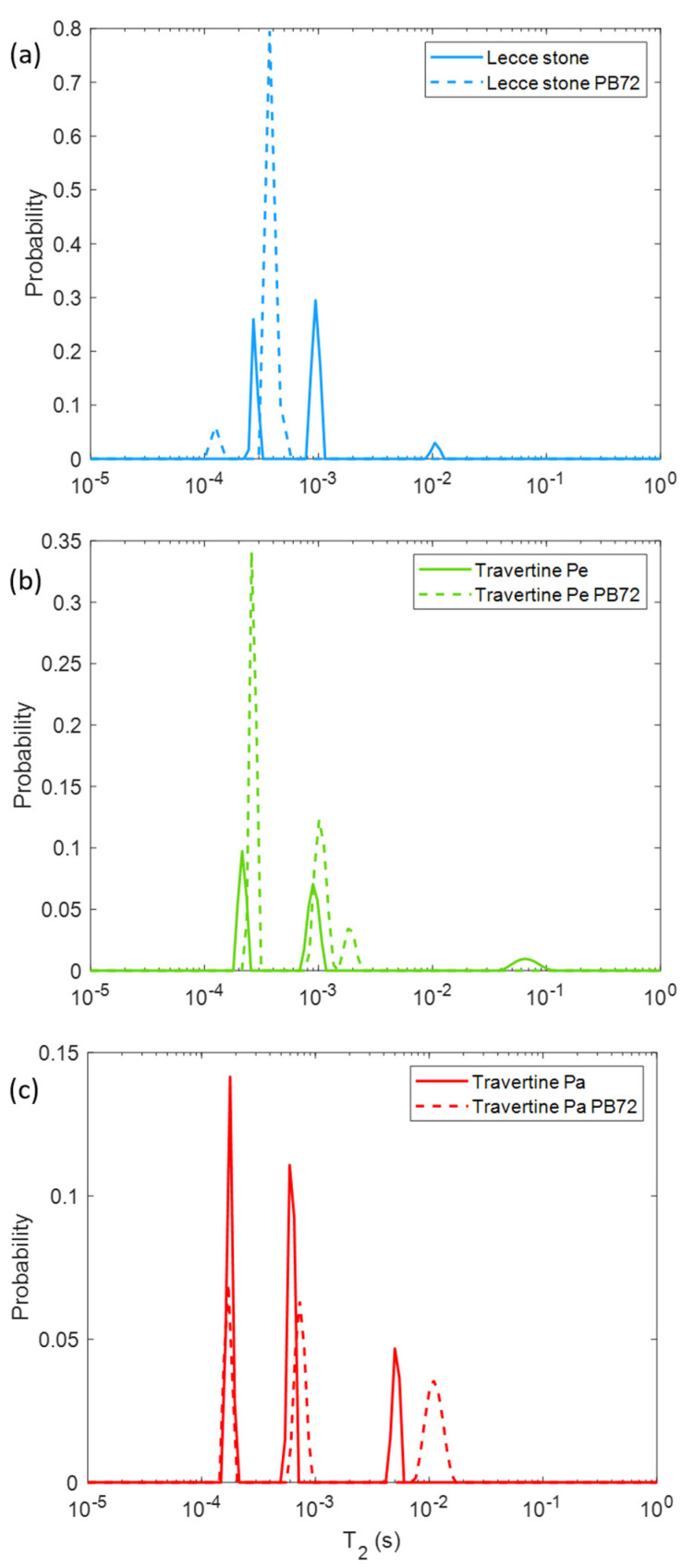
Transversal relaxation time (*T*_2_) distribution for (**a**) Lecce stone, (**b**) Travertine Pe, and (**c**) Travertine Pa before (solid lines) and after (dashed lines) the Paraloid B72 application.

**Figure 5 molecules-26-03697-f005:**
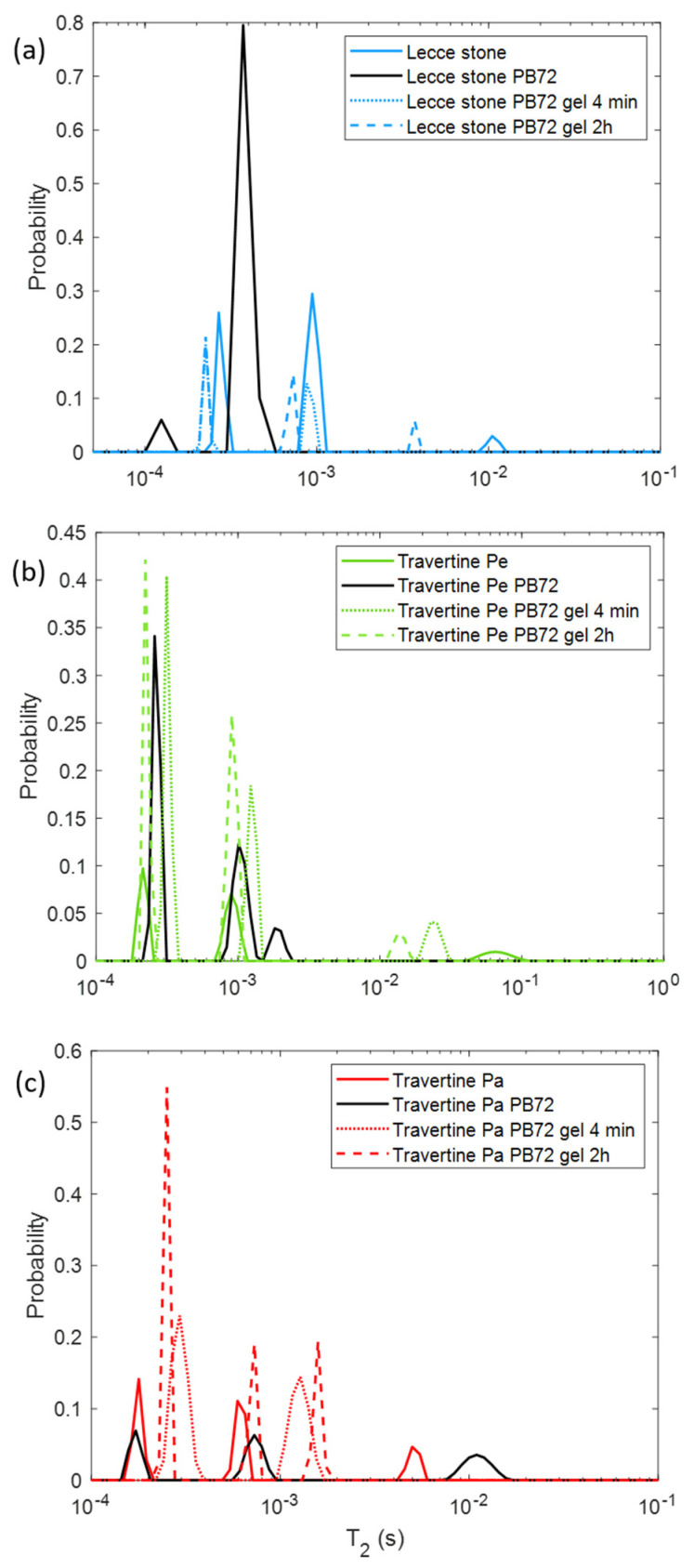
Transversal relaxation time (*T*_2_) distribution for (**a**) Lecce stone, (**b**) Travertine Pe, and (**c**) Travertine Pa before the treatment (solid lines), after PB72 application (black solid lines), and after the PVA-borax gel treatment with application times of 4 min (dotted lines) and 2 h (dashed lines).

**Figure 6 molecules-26-03697-f006:**
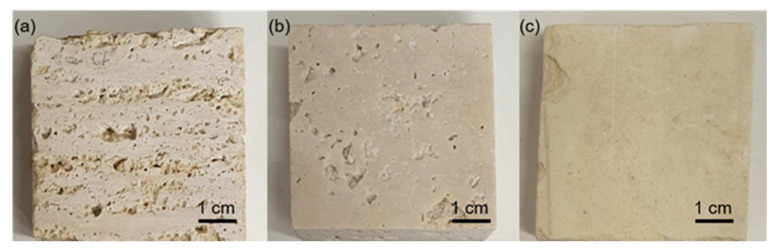
The three rock specimens used: (**a**) Travertine cut perpendicular to the bedding planes (Travertine Pe), (**b**) Travertine cut parallel to the bedding planes (Travertine Pa), and (**c**) Lecce stone.

**Figure 7 molecules-26-03697-f007:**
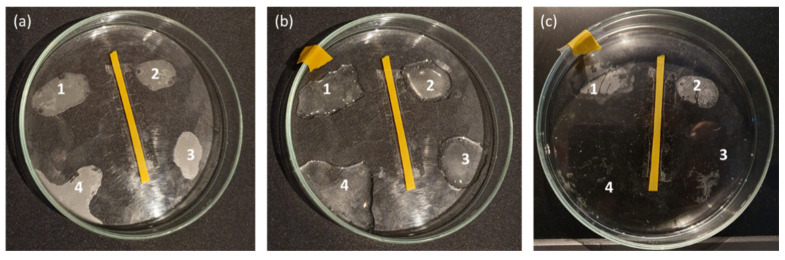
Preliminary evaluation of the exposure times needed for the removal of Paraloid B72 (2% in acetone *w*/*w*) by employing the PVA-B based HPVD. A glass Petri dish was employed for the experimental procedure. In (**a**) 4 stains (1, 2, 3, 4) of Paraloid B72 are shown. In (**b**), the same stains were treated with the PVA-PEO borax hydrogel, which was removed, respectively, 4 min (1), 30 min (2), 2 h (3), and 12 h (4) after its application. In (**c**) it is possible to appreciate the efficacy of the PVA-PEO borax hydrogel in the removal of the polymer over time, by observing the residues of Paraloid B72 on the glass.

**Figure 8 molecules-26-03697-f008:**
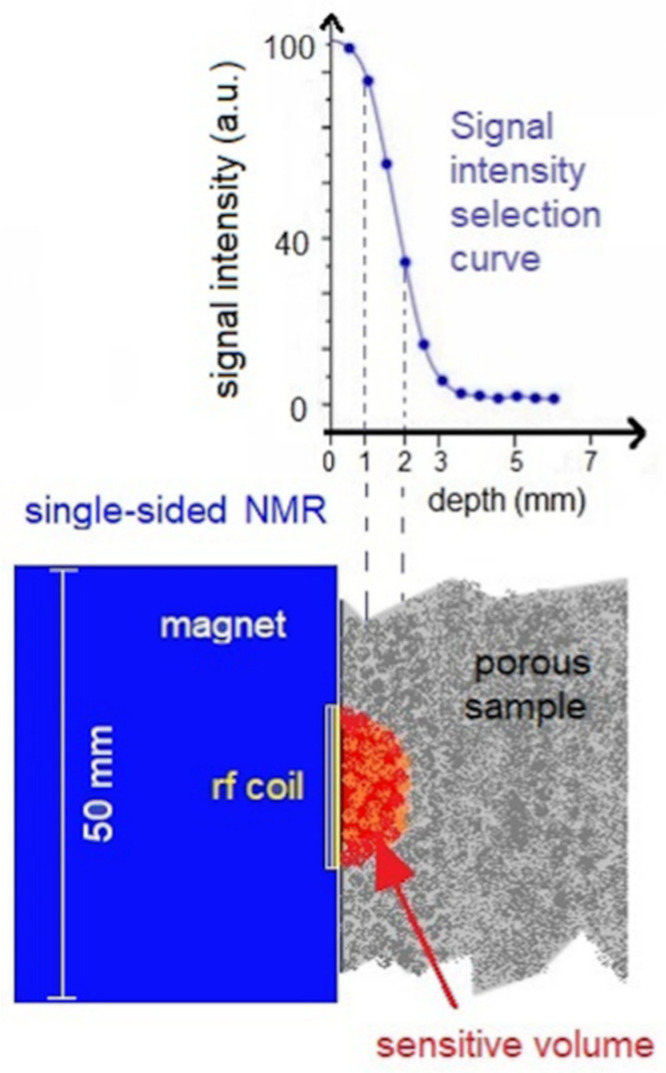
Schematic representation of the shape and the extension of the sensitive volume in the sample from which the NMR signal is obtained and *T*_2_ relaxation times are extracted.

**Figure 9 molecules-26-03697-f009:**
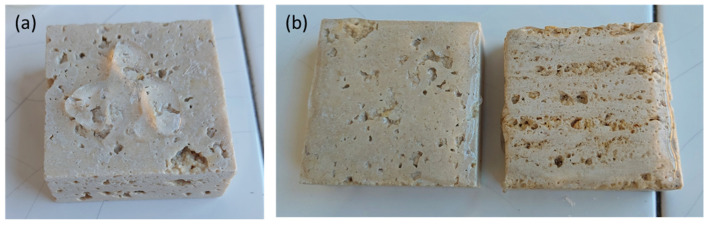
Application of the PVA-PEO borax hydrogel on the specimens. (**a**) A small quantity of gel applied on the stone surfaces. (**b**) The gel adapted to the shape of the samples, forming a homogeneous transparent film on the stone surfaces.

**Figure 10 molecules-26-03697-f010:**
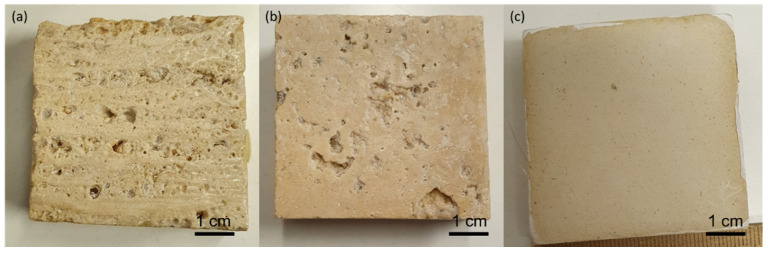
Appearance of (**a**) Travertine Pe cut perpendicular to the bedding planes, (**b**) Travertine Pa cut perpendicular to the bedding planes, and (**c**) Lecce stone after PB72 coating and the cleaning test with the PVA-based hydrogel for 2 h. Few solid residues of the hydrogel can be observed in (**a**,**b**).

**Table 1 molecules-26-03697-t001:** **CPMG** parameters for the acquisition of the *T*_2_-decay signal.

Parameters	Hydrogel	PB72	Stones
TR (ms)	2000	2000	500
TE (ms)	0.04	0.04	0.03
NS	512	2048	2048
Number of points	6500	80	200
First delay (ms)	0.04	0.04	0.03
Last delay (ms)	400	4	6

## Data Availability

The data presented in this study is openly available in Mendeley Data at 10.17632/867jftp6zz.1 [61].

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
