# Peer review of "Single-Sided Portable NMR Investigation to Assess and Monitor Cleaning Action of PVA-Borax Hydrogel in Travertine and Lecce Stone"

_molecules, 2021, doi:10.3390/molecules26123697_

Round 1

Reviewer 1 Report

The potential of PVA-borax hydrogel in cleaning limestones and the dependence of the cleaning on the rock’s porosity and on the action time of the hydrogel treatment has been assessed using NMR relaxometry technique adapted for non-destructive applications to cultural heritage. The distributions of transverse relaxation times T2 are quantified in samples of stones, which differed in origin, orientations of cuts, and specific treatment. The obtained T2 data suggest that the effectiveness of the cleaning strongly depends on the stone porosity.

Overall, the proposed manuscript reports many new findings and original observations obtained with novel physicochemical methods, which makes it interesting to a wide range of readers. The offered interpretation of the experimental results is logical and consistent and is well related to the results of previous studies. The language of the manuscript is clear and concise, and the preceding research is appropriately acknowledged. Both the topic of the work, the methods, and the object of research are well suited for the journal Molecules, and the manuscript can be published as-is.

Author Response

The authors would like to thank the Reviewers for their comments. We have addressed all the issues, and we feel the revised version of the manuscript has improved thanks to the changes that have been suggested. We hope the Reviewers will find this paper suitable for publication.

In order to facilitate the reading of the responses to the Reviewers’ comments, we report the original comments in bold, and the Authors’ replies in normal no-bold character.

The potential of PVA-borax hydrogel in cleaning limestones and the dependence of the cleaning on the rock’s porosity and on the action time of the hydrogel treatment has been assessed using NMR relaxometry technique adapted for non-destructive applications to cultural heritage. The distributions of transverse relaxation times T2 are quantified in samples of stones, which differed in origin, orientations of cuts, and specific treatment. The obtained T2 data suggest that the effectiveness of the cleaning strongly depends on the stone porosity.

Overall, the proposed manuscript reports many new findings and original observations obtained with novel physicochemical methods, which makes it interesting to a wide range of readers. The offered interpretation of the experimental results is logical and consistent and is well related to the results of previous studies. The language of the manuscript is clear and concise, and the preceding research is appropriately acknowledged. Both the topic of the work, the methods, and the object of research are well suited for the journal Molecules, and the manuscript can be published as-is.

Response: The authors thank the Reviewer for his/her positive comments on the manuscript.

Reviewer 2 Report

In this manuscript, authors report an investigation of the potential of PVA-borax hydrogel in cleaning limestones of different porosities by measuring NMR T2 relaxation times. In my opinion, the manuscript has significant shortcomings addressed below and is not suitable for publication in Molecules in the present form.

  1. The manuscript is hard to follow. There are many errors in grammar and clarity, and a native English speaker should edit the manuscript.
  2. Parts of the manuscript (e.g. Introduction) are too long and tedious to read.
  3. x-axes are not labeled in all Figures. In addition, it would be more convenient and clearer to report T2 data displayed in Figures in ms.
  4. It is not clear why the authors have chosen PVA-based gel application times of exactly 4 minutes and 2 hours?
  5. The authors derived their conclusions only based on NMR T2 relaxation times. They should use other complementary methods to confirm their conclusions.

Author Response

The authors would like to thank the Reviewers for their comments. We have addressed all the issues, and we feel the revised version of the manuscript has improved thanks to the changes that have been suggested. We hope the Reviewers will find this paper suitable for publication.

In order to facilitate the reading of the responses to the Reviewers’ comments, we report the original comments in bold, and the Authors’ replies in no-bold normal character.

In this manuscript, authors report an investigation of the potential of PVA-borax hydrogel in cleaning limestones of different porosities by measuring NMR T2 relaxation times. In my opinion, the manuscript has significant shortcomings addressed below and is not suitable for publication in Molecules in the present form.

Point 1: The manuscript is hard to follow. There are many errors in grammar and clarity, and a native English speaker should edit the manuscript.

Response 1: We thank the Reviewer for his/her comments. We asked to a native English speaker to edit our manuscript, we hope it is now clearer and easier to follow.

Point 2: Parts of the manuscript (e.g. Introduction) are too long and tedious to read.

Response 2: We also shortened and improved the manuscript introduction according to the Reviewer’s suggestion.

Point 3: x-axes are not labelled in all Figures. In addition, it would be more convenient and clearer to report T2 data displayed in Figures in ms.

Response 3: Regarding the label of x-axes, when more than one subplot is inserted in the same figure and these subplots have the same label for the x axis, the common practice to both matlab and python users is to set only one time the x-axis label. In this way the inserted x-axis label is referred to all the x-axes of all subplots and this also allows not to waste space. We agree that T2 can be displayed both in seconds or milliseconds. Anyway, it is more convenient to work with seconds when fits are performed by Inverse Laplace Transform algorithm.

Point 4: It is not clear why the authors have chosen PVA-based gel application times of exactly 4 minutes and 2 hours?

Response 4: We have chosen 4 minutes and 2 hours as gel application times because 4 minutes was the time suggested by Riedo et al. 2015 (https://doi.org/10.1186/s40494-015-0053-2), which is the reference we followed for the hydrogel preparation. The second application time of 2 hours was selected on the base of previous tests we performed on a glass beaker. These preliminary tests can be found in section 4.2.2 and they were performed in order to test the effectiveness of PVA-based gel in removing of Paraloid B72 layer with different application times. From these tests, when PVA-gel is applied for 2 hours it seems to better remove PB72 layer in respect to the application time of 4 minutes suggested by Riedo et al. 2015.

Point 5: The authors derived their conclusions only based on NMR T2 relaxation times. They should use other complementary methods to confirm their conclusions.

Response 5: the aim of this preliminary work was to test if the T2 parameter measured by portable NMR was sensitive to surface modifications during treatment with PB72 and PVA-gel and to differences in the porosity of the samples. To this end, we believe that NMR results we showed can answer to the abovementioned questions. Further tests and measurements with different NMR parameters and also complementary techniques will be performed in future works.

Reviewer 3 Report

The paper reports the understanding of cleaning process of limestone observed by NMR with single-sided magnet. Porosity and presence of paramagnetic impurities are evaluated by 1H T2 relaxation time though ILT spectra. Three different limestones with and without PB2 application are used before and after PVA treatment for cleaning. The detailed analysis of T2 relaxation time reveals that 1) PVA removes both paramagnetic impurities and PB72 in Lecce stone, 2) PVA is not effective in Travertine Pe, and 3) PVA eliminate paramagnetic impurities but not PB72 in Travertine Pa.

The topic of the research is relevant, the experiment details and discussions are clearly described, and the conclusion is well supported by the experimental observation. In general, I recommend the manuscript for publication, however, some re-arrangement of the paper is needed to improve the clarity.

Page8 line 197: The paragraph starting with “In the last years, …..[48, 49]” is rather introductive one. It should appear before Results section. This helps readers for better understanding.

Page3 line 103: “On the other hand” is mis-located as the discussion following doesn’t related to the evaluation method. This sentence can be placed at the end of the succeeding paragraph. (line 119).

Editorial corrections:

Line 123, 198: Cultural Heritage -> cultural heritage

Line 138: 561ms and 166ms -> 561 ms and 166 ms (insert space between number and unit)

Author Response

The authors would like to thank the Reviewers for their comments. We have addressed all the issues, and we feel the revised version of the manuscript has improved thanks to the changes that have been suggested. We hope the Reviewers will find this paper suitable for publication.

In order to facilitate the reading of the responses to the Reviewers’ comments, we report the original comments in bold, and the Authors’ replies in no-bold character.

The paper reports the understanding of cleaning process of limestone observed by NMR with single-sided magnet. Porosity and presence of paramagnetic impurities are evaluated by 1H T2 relaxation time though ILT spectra. Three different limestones with and without PB2 application are used before and after PVA treatment for cleaning. The detailed analysis of T2 relaxation time reveals that 1) PVA removes both paramagnetic impurities and PB72 in Lecce stone, 2) PVA is not effective in Travertine Pe, and 3) PVA eliminate paramagnetic impurities but not PB72 in Travertine Pa.

The topic of the research is relevant, the experiment details and discussions are clearly described, and the conclusion is well supported by the experimental observation. In general, I recommend the manuscript for publication, however, some re-arrangement of the paper is needed to improve the clarity.

Comments:

Page8 line 197: The paragraph starting with “In the last years, …..[48, 49]” is rather introductive one. It should appear before Results section. This helps readers for better understanding.

Page3 line 103: “On the other hand” is mis-located as the discussion following doesn’t related to the evaluation method. This sentence can be placed at the end of the succeeding paragraph. (line 119).

Editorial corrections: Line 123, 198: Cultural Heritage -> cultural heritage

Line 138: 561ms and 166ms -> 561 ms and 166 ms (insert space between number and unit).

Response: The authors thank the Reviewer for his/her positive comments on the manuscript. We asked to a native English speaker to edit our manuscript. We have now inserted the suggested corrections in the emended manuscript. However, regarding moving the paragraph starting with “In the last years, …..[48, 49]” at pag. 8, in the introduction, unfortunately we couldn't do it because another reviewer asked us to shorten the introduction.

Reviewer 4 Report

A portable NMR instrument was used to measure the T2 relaxation time distribution of protons in pristine Travertine and Lecce stone materials and the same materials treated with Paraloid B72, a PVA- 132 borax hydrogel, and Paraloid B72 treated materials covered with the PVA- 132 borax hydrogel.

The results obtained can be interpreted as evidence of the replacement of Paraloid B72 with the PVA-132 borax hydrogel on these stone materials. Neither the mechanism of this replacement, nor the interaction of the PVA- 132 borax hydrogel with these stone materials were studied.

The paper is well written and easy to read. I do not have any question about the quality of the manuscript. However, I doubt that the topic of this manuscript is interesting for the Molecules audience. MDPI journals Applied Sciences, Materials, Processes or Coatings would be much better suited for this manuscript.

Author Response

The authors would like to thank the Reviewers for their comments. We have addressed all the issues, and we feel the revised version of the manuscript has improved thanks to the changes that have been suggested. We hope the Reviewers will find this paper suitable for publication.

In order to facilitate the reading of the responses to the Reviewers’ comments, we report the original comments in bold, and the Authors’ replies in no-bold normal character.

A portable NMR instrument was used to measure the T2 relaxation time distribution of protons in pristine Travertine and Lecce stone materials and the same materials treated with Paraloid B72, a PVA- 132 borax hydrogel, and Paraloid B72 treated materials covered with the PVA- 132 borax hydrogel.

The results obtained can be interpreted as evidence of the replacement of Paraloid B72 with the PVA-132 borax hydrogel on these stone materials. Neither the mechanism of this replacement, nor the interaction of the PVA- 132 borax hydrogel with these stone materials were studied.

Comments: The paper is well written and easy to read. I do not have any question about the quality of the manuscript. However, I doubt that the topic of this manuscript is interesting for the Molecules audience. MDPI journals Applied Sciences, Materials, Processes or Coatings would be much better suited for this manuscript.

Response: The authors thank the Reviewer for his/her positive comments on the manuscript. We thank the Reviewer for his suggestion. However, we have chosen to submit our manuscript to be included in the collection of papers of the special issue of Molecules: “Physical Chemistry in Cultural Heritage”. The Guest Editors gave the follow Special Issue Information:

“Recently, the application of chemistry to the knowledge of works of art and archaeological objects has played an important role in the conservation and restoration of the developed world's cultural heritage. From this point of view, the investigation of physico-chemical processes involving both natural and voluntary surface modification of artifacts need to be considered, especially the study of degradation processes and the application of new materials for restoration…”.

Therefore, we think that our manuscript fits well to this special issue of Molecules. We asked to a native English speaker to edit our manuscript.

Round 2

Reviewer 2 Report

The revised version of the manuscript is suitable for publication in Molecules.